# Disparities of women's authorship in Colombia: A cross-sectional analysis

**María Alejandra Gutiérrez Torres**[1,2,*], **Silvana Ruiz**[1], **Karen Morales**[1], **Laura Rincon**[1], **Frans Serpa**[3], **Camila Gómez**[1], **Michelle M. Ahrens**[1], **Felipe Duran**[1], **Abul Ariza Manzano**[3], **Santiago Callegari**[1,4]

1 Faculty of Medicine, Universidad de Los Andes, Bogotá, Colombia, 2 Yale Child Study Center, Yale University, New Haven, Connecticut, United States of America, 3 Department of Cardiology, Beth Israel Deaconess Medical Center, Boston, Massachusetts, United States of America, 4 Vascular Medicine Outcomes Group, Yale University, New Haven, Connecticut, United States of America

* maria.a.gutierrez@yale.edu

## Abstract

Accepted manuscripts published in medical journals play a crucial role within the scientific community. Over the past few decades, there has been a gradual increase in the number of women entering the medical field. However, women remain under-represented as first and last authors in medical journals. This lack of representation makes it harder for them to reach leadership roles and advance academically. Even if this has been studied in high-income countries, low- and middle-income countries still lack evidence to prove this fundamental problem. This study aims to do this by investigating the gender distribution among authors and exploring disparities in authorship in Colombia. The analysis encompassed 6,088 articles derived from 54 research journals obtained from the official website of Colombia's Ministry of Health. The journals included enhance the significance of this paper, as they are typically not included in indexed databases. Consequently, their inclusion in gender evaluations has been limited in previous studies. These were predominantly original research articles, although case reports and reviews were also present. Until now, there has been no assessment of gender disparities in authorship within medical and surgical specialty journals in Colombia. The presence of enduring gender differences in medical authorship in Colombia remains evident, independent of the temporal, geographical, or academic domain. Even when factors such as medical specialization and geographic location influenced women's authorship proportions, the gap persisted in all cases. This highlights the critical need for increased support for women researchers and equitable resource allocation to address the specific medical specializations as well as geographical locations that we found were even more affected by these gender disparities. This paper highlights the urgent need to address gender disparities in the authorship of medical and surgical research publications in Colombia and other Latin American countries. Addressing these disparities is a critical step toward assisting women in advancing in an equal and fair medical profession.

**Data availability statement:** All relevant data are within the paper and its Supporting Information files.

**Funding:** The authors received no specific funding for this work.

**Competing interests:** The authors have declared that no competing interests exist.

## Introduction

Disparities in women's authorship within academic literature have been a subject of increasing concern across various medical and scientific disciplines. As Muric et al. highlight, gendered authorship disparities are undeniably present in medical academia, with lower proportions of publications by women authors in specific medical fields [1]. This is important not only because it highlights the gender disparities in medical academia but also because accepted manuscripts in medical journals are one of the primary forms of recognition within the scientific community. So, it plays a vital role in advancing careers in the healthcare professions [2,3]. Yet, this exacerbates the obstacles women face in their pursuit of success in the medical field, in contrast to those encountered by men.

While the underrepresentation of women as first and last authors in medical journals has been well documented, it is important to note that authorship practices can differ significantly across disciplines and publications. For example, disciplines like economics and physics frequently use alphabetical author ordering or include a credit author statement that explicitly details each author's contribution, making the first author appear random and unrelated to contribution levels. However, in the biological sciences, and thus in medical academia, the author list is frequently strictly ranked according to contribution. The researcher who primarily conducted the research is usually listed first, indicating their significant hands-on involvement and contribution to the project. The final position, often held by the principal investigator, recognizes their role in supervising the research. Given that this paper focuses on medicine, the traditional metric of first authorship is especially relevant and impactful for assessing career advancement and academic achievement.

This study uses the traditional metric of first authorship because it is widely used and has a recognized impact in medical academia, particularly in Colombia, where such conventions continue to play an important role in shaping academic careers. However, we recognize the limitations of using first authorship as the only indicator of gender disparities in academia. To provide a more complete picture, we also look at broader authorship patterns and contributions, such as last authorship, which frequently indicates seniority and scholarly authority.

Furthermore, the last author performs an important role in academic research and is traditionally reserved for senior researchers such as principal investigators or project leaders. This position, which frequently indicates a researcher's significant role and authority within the team, is more than just a symbolic gesture; it directly reflects an individual's standing and influence in their field. This strategic placement is critical for understanding career paths and mentorship dynamics in academia, especially when investigating the advancement of women in these roles. Disparities in last authorship can provide valuable insights into the structural challenges and potential barriers that women researchers face when advancing to senior levels. As a result, we purposefully selected women last authorship as an important outcome variable.

Filardo et al. [4] reported that the representation of women as first authors in high-impact medical journals seem to have stayed constant and even declined in some journals. Their findings revealed significant differences between journals in the likelihood of the first author of an original research article being women, indicating that the underrepresentation of women among the leaders of high-impact original research is an ongoing concern. This evidence underscores the enduring challenges women face in achieving equitable representation as authors in medical research publications despite advancements in medicine and the increasing participation of women.

The persistent underrepresentation of women in research, particularly in Latin America [4], emphasizes the importance of this study. Even if this is a problem that has affected uncountable women around the globe, these disparities have not received extensive evaluation

in most countries worldwide, especially in low- and middle-income countries, such as Latin American countries like Colombia [4]. According to recent UIS data, less than 30% of the world's researchers are women, highlighting the need for targeted interventions to address this imbalance. Furthermore, unequal resource allocation has been identified as a significant barrier to advancing women researchers in Latin America [5,6]. This study highlights the significant gaps in data regarding the scope and severity of gender inequality in scientific fields at the regional level. By identifying these gaps, the paper underscores the need for more targeted research to inform policy and practice effectively.

The participation of women in science in Colombia continues to be an unescapable issue [7]. As Lopez-Aguirre et al. show, there is a widespread lack of gender parity as well as an underrepresentation of women in science in Colombia across the twenty-first century and in science-related work fields [8,9]. This is why we need to emphasize the need for continued attention and action to address these imbalances, and the first step to do this is by recognizing the problem. This study aims to investigate gender distribution among authors and explore disparities in authorship in Colombia, providing a timely and essential contribution to understanding gender disparities in authorship within the Colombian academic landscape.

Social and cultural norms, a lack of mentorship opportunities, and individual perceptions of work-life balance are all likely contributing factors to the persistence of these disparities. Through a comprehensive review of existing literature and empirical data, the study will provide a deep understanding of the factors contributing to regional imbalances and the underrepresentation of women in research. This study highlights persistent gender disparities in first authorship in medical journals published in Colombia, identifying a clear pattern where women are less likely to be first authors. While this analysis does not directly establish causative factors or test specific interventions, it underscores the need for deeper investigation into the reasons behind these disparities. The findings serve as a preliminary step towards informing future experimental or quasi-experimental research that could evaluate the effectiveness of policy interventions aimed at reducing these gender gaps.

The current findings are expected to catalyze discussions among policymakers, funding agencies, and research institutions regarding the necessity of targeted studies to understand and address the root causes of gender inequity in academic authorship. It is hoped that this will eventually lead to collaborative efforts to craft and implement evidence-based policies that enhance women's representation and inclusivity in the scientific community across Latin American nations.

While this study does not propose specific interventions, it brings to light significant trends that warrant further exploration. By documenting these disparities, it lays the groundwork for future research that could lead to more effective solutions. In this way, the study contributes to ongoing efforts to support women researchers and to distribute resources more equitably, striving towards a more just and supportive research environment for all scientists in the region.

The anticipated results of this study are expected to stimulate collaborative efforts among policymakers, funding agencies, and research institutions to increase women's representation and inclusivity in research throughout Latin American nations. By shedding light on the existing disparities and proposing evidence-based solutions, this study seeks to contribute to the creation of a fairer and more equitable academic environment in the region.

In conclusion, this study represents a critical step towards addressing the urgent need for enhanced support for women researchers and equitable resource allocation in Latin America. By promoting gender equity and inclusivity in research, the study aims to contribute to the creation of a more just and supportive research environment for all researchers.

## Methods

### Criteria for journal selection

Journals in the biomedical sciences related to medicine that have been continuously published for two years or more were defined as inclusion criteria. Our analysis included both university-affiliated medical journals with a broad scope and specialty-specific medical journals. Journals in biological or healthcare sciences that did not primarily focus on medical sciences, such as veterinary science, biology, nursing, and dentistry, were excluded.

### Journal search methodology

Scielo, Directory of Open Access Journals, PubMed, and Publindex were screened to include all possible Colombian journals. Given the limited number of Colombian journals in databases such as PubMed, we conducted a deeper search in order to ensure inclusivity, taking into account journals from various regions and specialties within Colombia. Publindex, a government-affiliated entity overseeing scientific journals in Colombia, was our primary source for identifying Colombian journals. We excluded journals not classified as medical sciences and any journals with titles indicating a focus other than medicine, such as biological sciences (e.g., veterinary, biology) or healthcare sciences (e.g., nursing, dentistry). A second author independently conducted the same protocol to ensure the rigor of our screening process, and only journals that received consensus from both authors were included in our analysis. Following that, we reviewed all selected journals to ensure they focused exclusively on medicine. The journal in question was excluded if any articles unrelated to medicine were identified during this review (Fig 1).

### Criteria for article selection

Articles were selected in accordance with the following inclusion criteria: 1) original research articles, review articles, and case studies; and 2) online public articles published in Colombian journals between 2018 and 2022. We excluded editorials, opinion pieces, special editions, and unaffiliated articles as they were not considered to be within the scope of our research. Furthermore, we excluded journals that included articles from multiple specialties to avoid bias.

### Article abstractions

We conducted a cross-sectional study to determine authorship rates across multiple medical and surgical specialties. To accomplish this, we performed a bibliographic search that included all original articles published between January 20, 2018, and December 14, 2022, specifically in journals affiliated with Colombia's leading medical associations.

We gathered information from the websites of each journal from January 2023 to May 2023, and we collected data from each journal's website. Afterward, we used the online Genderize webpage/application (https://genderize.io/) to determine the gender of the authors based on the names of the first and senior authors. Furthermore, we conducted Google searches for the authors' names when required, cross-referencing the information with the official websites of their institutions. The software determines the gender of each name using a probabilistic method based on global published literature. Two or more researchers independently assessed the data at each stage, and consensus resolved discrepancies. When there was doubt about the gender of the authors, we excluded the article from the statistical analysis.

After excluding articles, the researchers independently collected data from the remaining articles. The following data was extracted during this data collection process: article type, first author's name, last author's name, corresponding author's name, and affiliations. After that,

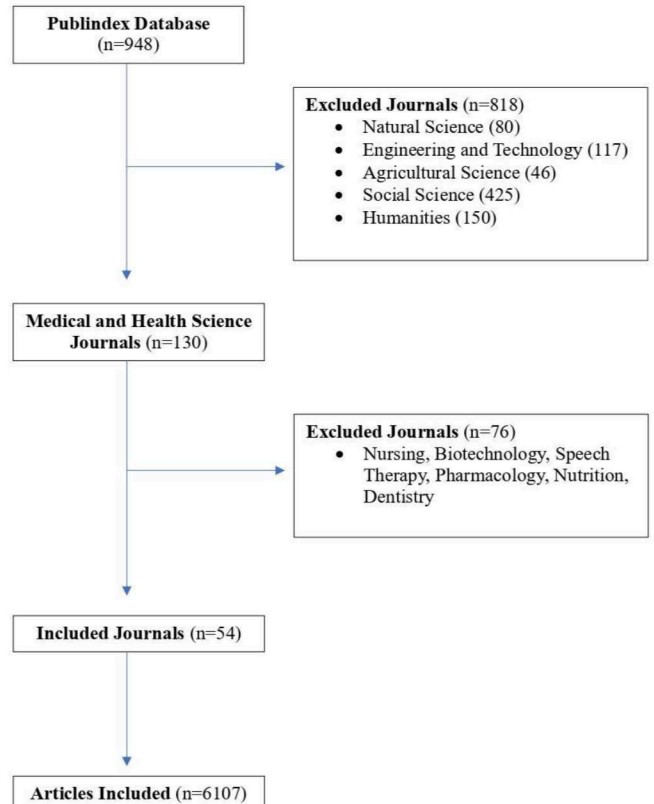

**Fig 1. Flowchart of including and excluding scientific journals and articles for the analysis.**

all collected data was consolidated and saved in a shared Microsoft Excel database. We used Fisher's exact test for our statistical analysis and calculated association measures using STATA 17 BE software. The level of significance we chose was 5%.

## Statistical analysis

All analyses were performed in a complete case dataset for the first and last authorship positions. Variables were summarized with frequencies of women and men authors for each type of article. Frequencies were compared using Fisher's exact test. Temporal trends were assessed graphically by plotting the proportion of women first and last authorship positions per year. Subsequently, articles were separated by the specialty of the journal and compared graphically using bar charts. Lastly, the proportion of women articles was also divided by the state of the author's affiliation and analyzed graphically using a heat map based on Colombia's political map.

In a separate analysis, we performed univariate and 2 multivariate logistic regression to explore the factors associated with women being (1) first authors and (2) last authors. The univariate logistic regression models evaluated the association of women's first authorship and journal, gender of the last author, medical specialty of the journal, type of article, city of affiliation of the first author, and city of affiliation of the last author. Similar analysis was performed for the last author. Lastly, our 2 logistic regression multivariate analysis models included gender of the last author (only included in model 1), gender of the first author (only included in model 2), medical specialty of the journal, type of article, city of affiliation of the first author,

and city of affiliation of the last author. We hypothesize that a last women author, medical specialties, and an affiliation located in the biggest cities in Colombia (Bogota and Medellin) will be associated with a women-first author. Similarly, we hypothesize that a first woman author and an affiliation located in the major cities of Colombia (Bogota and Medellin) will be associated with a woman's last authorship.

All analyses were performed using STATA 18 BE software. All tests were two-tailed, and a p-value <0.05 was considered statistically significant.

## Results

This study reviewed 54 journals obtained from the official website of Colombia's Ministry of Health, specifically from the journal regulation information available on publindex.com. The exclusion criteria were primarily focused on journals with broad coverage in biological or healthcare sciences (Fig 1). In total, 6,088 articles were included in our analysis. These were primarily original research articles, but there were also case studies and reviews, as shown in Table 1. However, the investigation was explicitly focused on research articles, considering that they have been extensively studied in the existing literature.

### Women's authorship over time

In terms of author gender distribution, data collected indicates that women author participation has generally remained stable over time. The articles were organized chronologically based on their publication year, showing that the number of women-first authors have steadily increased over the last five years. However, the number of women last authors has remained relatively constant during this time. Similar patterns were observed when we analyzed reviews and case reports by publication timeline, as detailed in Tables 1 and 2. Nonetheless, as illustrated in Fig 2, the differences between the years analyzed are not substantial enough on a year-by-year basis to suggest that time serves as a significant covariate in determining gender inequality. However, this may indicate a changing trend in the last few years.

**Colombian women authorship rates vary by specialty.** The journals were classified according to their primary focus, whether it was medicine, surgery, or a broader range of topics (miscellaneous). We also divided the journals into subspecialties, such as cardiology, nephrology, and plastic surgery, among others. When we studied research articles, we discovered significant differences in women author participation across journals, as shown in Fig 3. In general terms, surgical journals had fewer women authors than non-surgical journals, which is aligned with previous studies [10,11]. This trend was especially noticeable in orthopedic surgery, consistent with existing literature [12].

Surgical journals, particularly in orthopedics and neurosurgery, consistently showed fewer women authors compared to non-surgical journals like nutrition, dermatology, radiology, pediatrics, and public health. This aligns with existing literature and is clearly depicted in Fig 3.

**Table 1. Frequency of the first author being a woman when the last author is a woman.**

| Authorship between first and last authors according to the type of article | | | | | | |
|---|---|---|---|---|---|---|
| Type of article | Last women author | Last men author | p-value | First women author | First men author | p-value |
| Research article | 1322/3261 [**0.41**] | 1939/3261 [**0.59**] | <0.001 | 1504/3261 [**0.46**] | 1757/3261 [**0.54**] | <0.001 |
| Case Report | 569/1572 [**0.36**] | 1003/1572 [**0.64**] | <0.001 | 603/1572 [**0.38**] | 969/1572 [**0.62**] | <0.001 |
| Review Article | 535/1254 [**0.43**] | 719/1254 [**0.57**] | <0.001 | 573/1254 [**0.46**] | 681/1254 [**0.54**] | <0.001 |

**Table 2. Factors associated with the first author being a woman from 2018–2022 in Colombia.**

| Factors associated with the first author being a woman from 2018–2022 in Colombia | | OR | 95% confidence interval | p-value |
|---|---|---|---|---|
| Last author being women | | 1.55 | 1.39–1.73 | <0.001 |
| Subspeciality | Others | Reference | | |
| | Cardiology | 0.63 | 0.49–0.80 | <0.001 |
| | Gastroenterology | 0.34 | 0.26–0.45 | <0.001 |
| | Anesthesiology | 0.67 | 0.46–0.98 | 0.036 |
| | Dermatology | 2.44 | 1.62–3.68 | <0.001 |
| | Infectiology | 1.15 | 0.86–1.54 | 0.339 |
| | Neurology | 0.79 | 0.57–1.08 | 0.137 |
| | Hematology | 0.79 | 0.56–1.12 | 0.191 |
| | Public Health | 1.04 | 0.85–1.27 | 0.714 |
| | Radiology | 1.47 | 0.97–2.22 | 0.071 |
| | Sports Medicine and Rehabilitation | 0.64 | 0.44–0.93 | 0.02 |
| | Pediatrics | 0.92 | 0.65–1.29 | 0.614 |
| | Gynecology | 0.81 | 0.54–1.20 | 0.286 |
| | Urology | 0.75 | 0.55–1.02 | 0.066 |
| | Otorhinolaryngology [ENT] | 0.87 | 0.59–1.28 | 0.49 |
| | Neurosurgery | 0.27 | 0.15–0.49 | <0.001 |
| | Orthopedics/Traumatology | 0.1 | 0.06–0.18 | <0.001 |
| | Psychiatry | 0.88 | 0.61–1.29 | 0.525 |
| | Endocrinology | 0.62 | 0.45–0.86 | 0.004 |
| | Nephrology | 0.47 | 0.31–0.72 | 0.001 |
| | Critical Care | 0.56 | 0.40–0.78 | 0.001 |
| | Surgery | 0.34 | 0.22–0.52 | <0.001 |
| Year | 2018 | Reference | | |
| | 2019 | 1.07 | 0.90–1.27 | 0.461 |
| | 2020 | 0.97 | 0.82–1.14 | 0.701 |
| | 2021 | 0.95 | 0.80–1.11 | 0.499 |
| | 2022 | 1.18 | 1.00–1.39 | 0.056 |
| Type of article | Original Research | Reference | | |
| | Case Report | 0.73 | 0.64–0.84 | <0.001 |
| | Review | 0.97 | 0.84–1.11 | 0.648 |
| City | Bogotá | Reference | | |
| | Bucaramanga | 1.37 | 1.08–1.73 | 0.008 |
| | Medellín | 1 | 0.86–1.17 | 0.975 |
| | Cali | 0.87 | 0.70–1.07 | 0.187 |
| | International | 0.77 | 0.60–0.98 | 0.033 |
| | Other | 0.97 | 0.84–1.11 | 0.653 |

It should be noted that while the data presented in Fig 3 suggest heterogeneity in the proportion of women first authors across sub-specialties, gender-based selection into specific fields is a variable that could directly impact the number of women per sub-specialty that could be authors given the number of women sub-specialties. However, without detailed demographic data on the gender composition of each medical specialty in Colombia, our study is limited in its ability to fully understand whether the observed disparities in authorship are due to fewer women authors in surgical specialties or fewer women in these fields. This distinction is critical for accurately determining the nature of gender disparities in medical academia.

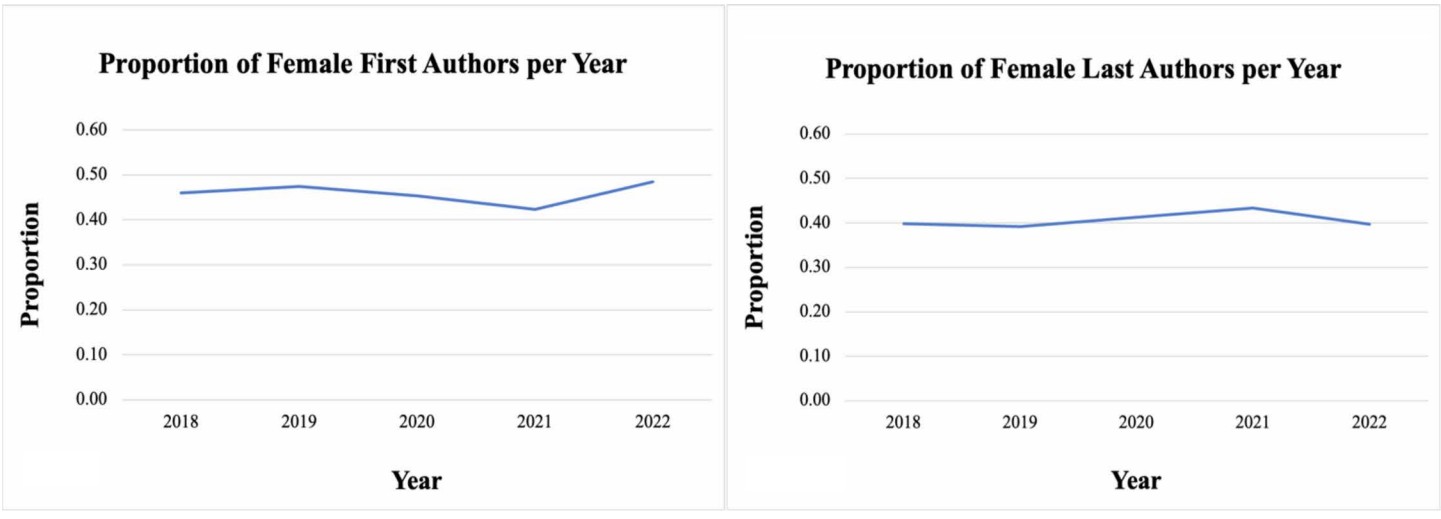

**Fig 2. Proportion of women first (A) and last (B) authors per year.**

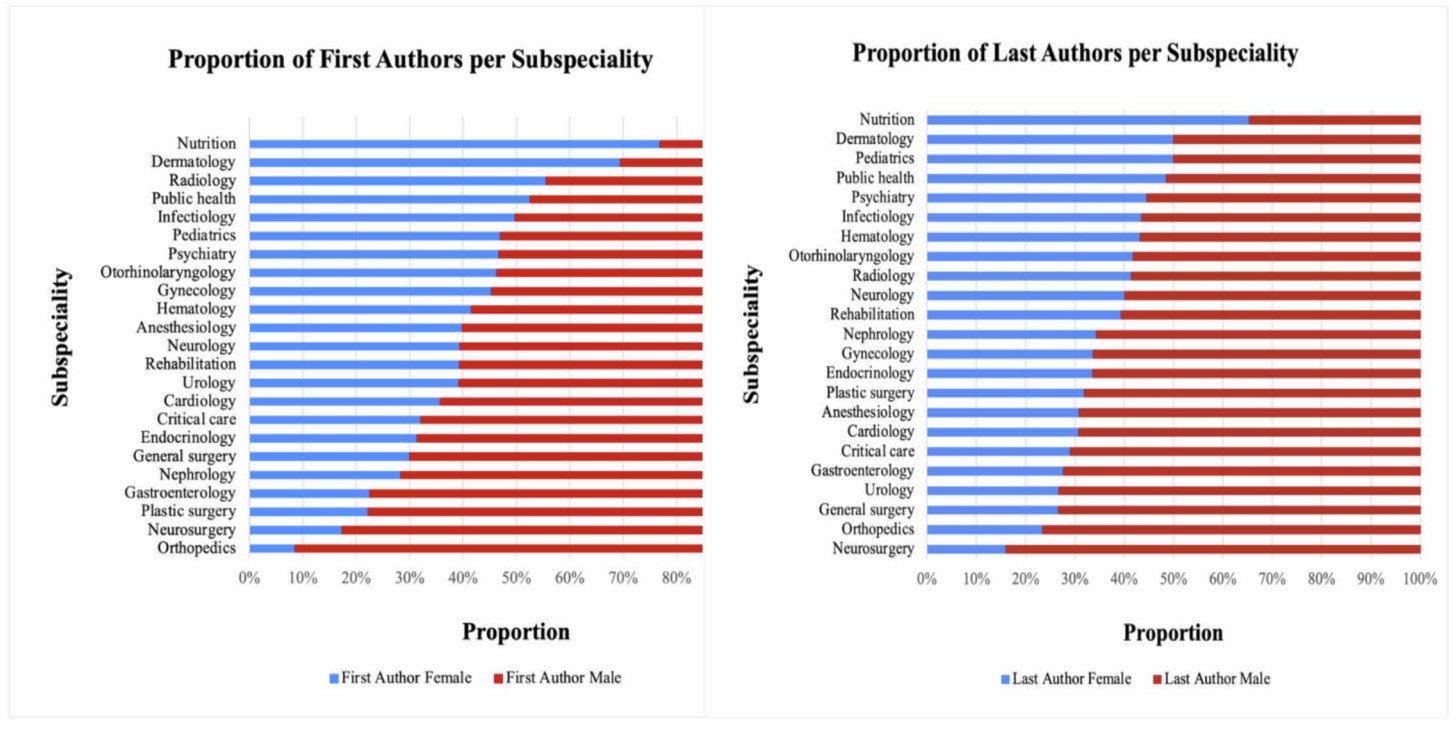

**Fig 3. Proportion of women first (A) and last (B) authors per subspecialty.**

Addressing the issue of gender disparities in authorship roles, it is imperative to bridge this gap and enhance the reliability of future research. The creation of a comprehensive database detailing the exact number of women professionals in each medical specialty is a crucial step. This database would enable researchers to account for the proportion of women in each specialty, thereby providing a clearer understanding of whether disparities in authorship roles per

sub-specialty are primarily driven by systemic biases or by the distribution of women across different fields. The ability to accurately measure the number of women in each specialty with first and last authorship is pivotal for developing precise and effective interventions.

Emphasizing the creation of this database promises to reduce biases in analyzing gender disparities and help develop informed and equitable policies. By understanding the unique dynamics of each specialty, interventions can be more precisely targeted to support women medical professionals, making gender equity in authorship a more attainable and measurable goal.

**Colombian regions exhibit similar trends.** Furthermore, despite varying prevalence rates, the trend of lower women authorship remained consistent across Colombian regions. As shown in Fig 2, the overwhelming majority of articles originated from Bogotá, Colombia's capital city. We identified five primary states that had a higher number of publication articles: Bogotá, Medellin, Cali, Bucaramanga, and Barranquilla, which include most of the country's largest cities. It can be evidenced in Fig 4 that there is an evident concentration of publications in the country's central areas. It is also worth noting that the data revealed a more significant proportion of states with women first authors than women last authors.

However, an intriguing pattern showed that although these main cities had higher publication patterns, they were not included in the states with higher women authors. According to Fig 4A, the proportion of women-first authors was higher in regions such as Casanare, Santander, Caquetá, and Nariño. In contrast, Fig 4B illustrates the distribution of women's last authors, showing a higher representation in Boyacá and Magdalena.

## Factors associated with women's first and last authorships

On the other hand, we calculated odds ratios to understand the factors associated with the first and last authors being women, as shown in Tables 2 and 3, adjusted by type of

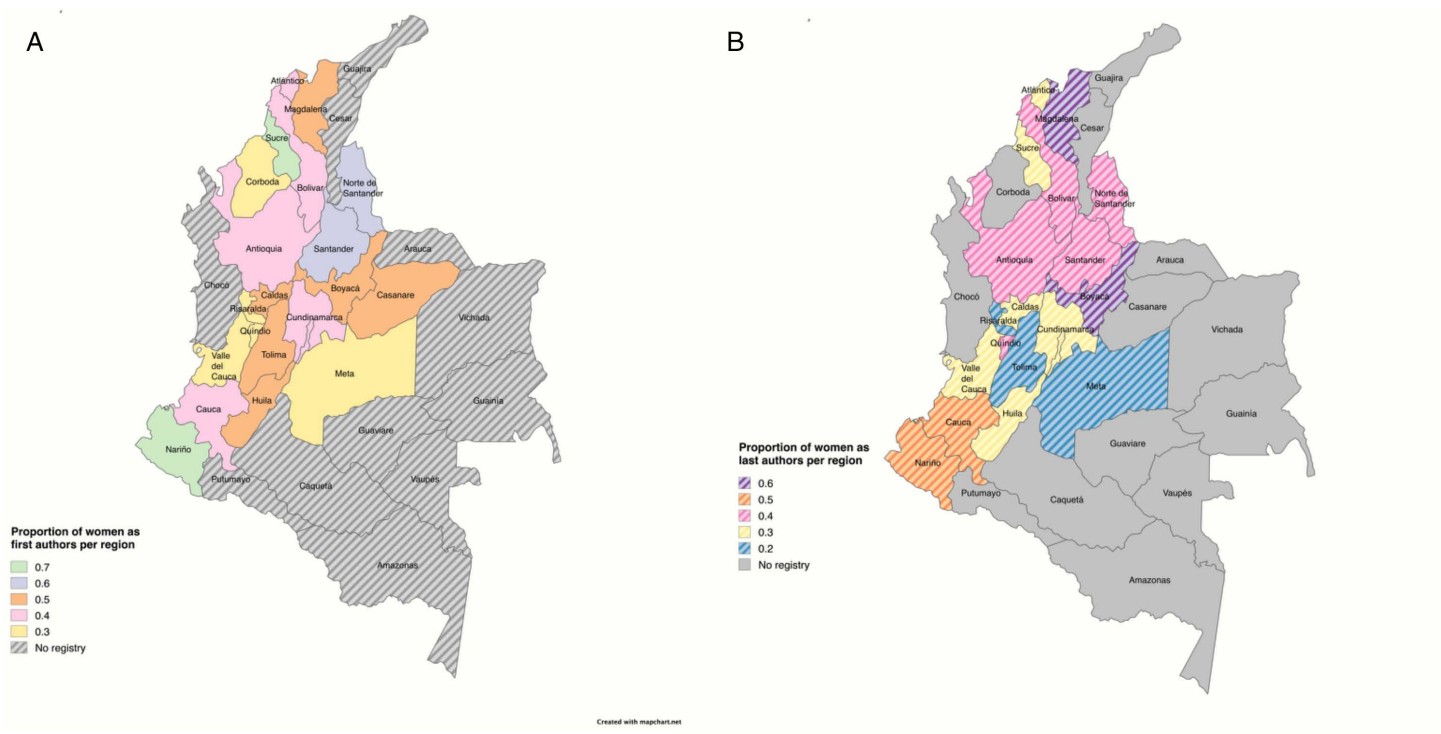

**Fig 4. Proportion of women first (A) and last (B) authors per region.**

**Table 3. Factors associated with the last author being a woman from 2018–2022 in Colombia.**

| Factors associated with the last author being a woman from 2018–2022 in Colombia | | OR | 95% confidence interval | p-value |
|---|---|---|---|---|
| First author being women | | 1.56 | 1.40–1.73 | <0.001 |
| Subspeciality | Others | Reference | | |
| | Cardiology | 0.58 | 0.45–0.74 | <0.001 |
| | Gastroenterology | 0.56 | 0.43–0.72 | <0.001 |
| | Anesthesiology | 0.58 | 0.39–0.85 | 0.006 |
| | Dermatology | 1.29 | 0.88–1.89 | 0.196 |
| | Infectiology | 1.05 | 0.79–1.41 | 0.717 |
| | Neurology | 0.83 | 0.60–1.14 | 0.251 |
| | Hematology | 1.04 | 0.73–1.47 | 0.835 |
| | Public Health | 0.99 | 0.81–1.21 | 0.927 |
| | Radiology | 0.89 | 0.59–1.36 | 0.597 |
| | Sports Medicine and Rehabilitation | 0.83 | 0.57–1.21 | 0.332 |
| | Pediatrics | 1.34 | 0.96–1.88 | 0.087 |
| | Gynecology | 0.68 | 0.45–1.03 | 0.068 |
| | Urology | 0.5 | 0.35–0.70 | <0.001 |
| | Otorhinolaryngology [ENT] | 0.89 | 0.60–1.31 | 0.547 |
| | Neurosurgery | 0.29 | 0.16–0.54 | <0.001 |
| | Orthopedics/ Traumatology | 0.4 | 0.27–0.59 | <0.001 |
| | Psychiatry | 0.89 | 0.61–1.30 | 0.548 |
| | Endocrinology | 0.65 | 0.46–0.91 | 0.012 |
| | Nephrology | 0.65 | 0.43–0.99 | 0.044 |
| | Critical Care | 0.55 | 0.39–0.78 | 0.001 |
| | Surgery | 0.57 | 0.38–0.87 | 0.008 |
| Year | 2018 | Reference | | |
| | 2019 | 1 | 0.84–1.20 | 0.957 |
| | 2020 | 1.12 | 0.94–1.32 | 0.197 |
| | 2021 | 1.19 | 1.01–1.40 | 0.039 |
| | 2022 | 0.97 | 0.82–1.15 | 0.697 |
| Type of article | Original Research | Reference | | |
| | Case Report | 0.88 | 0.77–1.01 | 0.065 |
| | Review | 1.11 | 0.97–1.27 | 0.14 |
| City | Bogotá | Reference | | |
| | Bucaramanga | 0.96 | 0.76–1.21 | 0.732 |
| | Medellín | 1.06 | 0.91–1.23 | 0.488 |
| | Cali | 0.8 | 0.64–0.99 | 0.04 |
| | International | 0.82 | 0.65–1.03 | 0.085 |
| | Other | 1.05 | 0.91–1.21 | 0.51 |

article, specialty, and geographical region. Factors positively associated with a woman being the first authors were dermatology articles and authors located in Bucaramanga, while factors negatively associated included most surgically related subspecialties, cardiology, gastroenterology, sports medicine, nephrology, and critical care-related articles.

When assessing factors associated with women being the last authors, the variable positively associated with this was the first author being a woman. The factors negatively associated were similar to the ones found for first authorship.

## Discussion

According to the UNESCO Institute for Statistics (UIS), the proportion of women in the global science workforce is only one-third. Nonetheless, Central Asia and Latin America are international pioneers in achieving gender equality in science at a regional level, with 48% and 45% rates, respectively [6]. However, despite all these efforts and advances, the lack of information on the extent and magnitude of gender disparity in science at local scales, particularly in nations with low research and development expenditure, poses a significant obstacle to the implementation of policies that seek to promote gender parity [13]. It has even been estimated that women's participation in low and middle-income countries such as Colombia and Argentina will face a decrease in women's participation in scientific publishing, creating a greater gap in gender parity [14].

In Colombia, gender disparities in authorship persist across various academic disciplines, regardless of time or region. In this nation, as in many others, the representation of women in the field of science is still lacking, as they only constitute 38% of researchers and 14% of active members in the Colombian Academy of Exact, Physical, and Natural Sciences [13]. In this country, women's authorship has remained stable over time, with surgical journals having fewer women authors than medical journals [15]. However, there is still a lack of literature that evidences these inequities in Colombia, which is the first step for reducing the authorship gap. This study aims to significantly enhance existing literature by including a wide range of medical and surgical specialties and non-PubMed indexed journals commonly found in Colombia.

After analyzing more than 6,000 articles, it has been evidenced that there are not only gender disparities in Colombia but also that these authorship trends differ in various geographic areas of the country. According to a study conducted in Colombia, only 12 out of the 32 Colombian states demonstrated a significant volume of academic publications in the medical field [3]. As shown, in Colombia, medical articles are not published at the same rate in every region, which may explain in some way why, in Fig 4, the center of the country exhibits a massive difference in publication rates compared to the peripherical states. However, among the states that do have high publication rates, only 5 regions out of 32, evidence a predominance of women authors. Nonetheless, as shown in Table 3, Cali has lower odds of having women as last authors compared to Bogotá, Colombia's capital city, which further attenuates in smaller cities and rural areas of the country.

When a country as complex as Colombia is analyzed, the cultural, economic, and social factors cannot be obviated, starting from the fact that they vary considerably. However, when Fig 4 is compared to what the Colombian researcher Sánchez-Torres has found in his studies. It can be evidenced that in most cases, the publication rates in the Colombian states have a direct relationship with each state's Gini coefficient [16]. This evidence shows the inequality that has to be addressed in Colombia regarding medical research, which may be one of the multiple underlying causes contributing to these disparities. For instance, in Colombia, the combination of limited funding and suboptimal working conditions has also contributed to Colombia having one of the lowest rates of doctoral graduates in Latin America (8 per million habitants) and a significant number of Colombian doctoral graduates residing overseas [17].

Even though the UNESCO Institute for Statistics (UIS) has declared that Latin American countries are world leaders in gender parity [6], according to our findings, women authorship has remained stable in Colombia. As López-Aguirre has sustained, medical and health science is the only field showing a temporal decrease in women's representation, losing 4.94% of representation between 2005 and 2015 [13]. This bias may relate to the fact that even though the gender disparity in authorship has been widely published, the majority of the studies are from

journals in the United States. There is minimal available literature in Latin America, with only two studies identified. Graner *et al.* conducted one in Brazil and discovered that there is an underrepresentation of women in surgical Brazilian journals [5]. With similar results, Dominguez *et al.* examined journals from Argentina and Chile, showing that women's authorship has increased over time, but they still present a considerable underrepresentation, especially as the last authors [18]. Similarly, post-pandemic global research in scientific publications, including the medical field, revealed that Latin America had the lowest percentage of women authorship, with 16% when compared to Asia, Africa, Europe, and Oceania [1].

However, it must be noted that while first authorship has increased slightly over the last five years, last authorship has remained consistent across all article types [19]. She also uncovered a higher presence of women first authors in articles where the last author was a woman, which is analogous to our results shown in Tables 2 and 3.

Another finding reveals the differences not only in the Colombian regions but also in the medical specialties that the journals have. This has been evidenced in previous studies, which have shown that the increase in women's representation in surgical journals has been less pronounced historically. In accordance with the described trends, women's representation in specific journals, such as Liver Transplantation, European Journal of Cardiothoracic Surgery, and Journal of Vascular Surgery, was significantly lower in the last authorships. For instance, the number of women's last authors increased at roughly half the rate of women's first authors in high-impact surgical journals, as Hart et al. discovered [20]. This is reflected as well in Tables 2 and 3, which indicate that women are more likely to be the first and last authors in subspecialties such as dermatology compared to neurosurgery, orthopedics, and traumatology, as well as in other fields such as gastroenterology and cardiology.

Despite the slow but steady increase in women authorship worldwide, it is critical to recognize that low and middle-income countries face significant challenges [4]. This emphasizes the importance of addressing the challenges that women face in this field and working toward gender parity. For instance, it is fundamental to enhance mentorship to promote women's professional development in academia across the region [21]. Also, considering what was evidenced in Colombia, future research should investigate whether similar gender disparities in authorship exist in other Latin American countries, as well as the factors that contribute to these disparities. This could be the first step to decreasing gender disparities in medical authorships.

Gender biases, cultural norms, and unequal access to education and career advancement may all contribute to the underrepresentation of women authors in Latin America. Eliminating these barriers and implementing gender-equitable policies is crucial for fostering an inclusive research environment [22]. This effort could include programs like women's mentorship, promoting work-life balance, and enforcing policies that ensure equal career opportunities. In addition, concerted efforts must be made to empower and support researchers in underrepresented regions by supplying them with the necessary resources and infrastructure for meaningful research engagement. This assistance could originate in the form of establishing research centers or partnerships in these areas, as well as facilitating networking opportunities to improve knowledge exchange.

In conclusion, regardless of timeframe, location, or field of study, our study reveals a persistent gender disparity in authorship in Colombia. This highlights the critical need for increased support for women researchers as well as equitable resource allocation to correct regional imbalances. Our comprehensive analysis offers practical recommendations and significantly contributes to ongoing efforts to create a more equitable research landscape. Despite its limitations, our study enriches the existing body of written work. It promotes collaborative efforts toward developing an inclusive research environment that recognizes and values the contributions of all researchers, regardless of gender or ethnicity.

## Study limitations

We acknowledge the inherent complexity of our research problem and the limitations of our investigation. First and foremost, because we used a manual article selection process, it is critical to recognize the possibility of human error. However, the large number of articles included in our study increases our confidence in the validity of our findings. Second, we needed to exclude specific journals outside the medical scope. However, deliberate efforts were made to ensure representation from key regions in Colombia, thereby increasing the depth of our research. Although our temporal analyses did not reveal any significant deviations, we must consider the potential impact of the COVID-19 pandemic on evolving trends. While our study was limited to Colombia, it is critical to recognize the need for similar initiatives in fields beyond our research boundaries. Thirdly, our statistical analysis is limited due to the variables available in Colombia. Some variables, such as the number of authors and women enrolled in the different subspecialties, might also play a role and will be addressed in future research efforts. Finally, we recognize that some of the journals we investigated may have needed more coverage of specific subspecialties, which could introduce bias. Nonetheless, as previously demonstrated, including subspecialty-focused journals provides a more accurate picture of women authorship trends. This aspect distinguishes our approach, especially when compared to academic publications primarily focused on educational contexts rather than clinical settings.

## Supporting information

**S1 Data. Database.**
(XLSX)

## Acknowledgments

Dr. Martha Gulati for their guidance and support on this project.

## Author contributions

**Conceptualization:** María Alejandra Gutiérrez Torres, Silvana Ruiz, Abul Ariza Manzano, Santiago Callegari, Felipe Duran.

**Data curation:** María Alejandra Gutiérrez Torres, Silvana Ruiz, Karen Morales, Laura Rincon, Michelle M. Ahrens, Abul Ariza Manzano, Santiago Callegari.

**Formal analysis:** María Alejandra Gutiérrez Torres, Silvana Ruiz, Karen Morales, Michelle M. Ahrens, Abul Ariza Manzano, Santiago Callegari.

**Funding acquisition:** María Alejandra Gutiérrez Torres.

**Investigation:** María Alejandra Gutiérrez Torres, Silvana Ruiz, Karen Morales, Laura Rincon, Frans Serpa, Michelle M. Ahrens, Abul Ariza Manzano, Santiago Callegari, Camila Gómez, Felipe Duran.

**Methodology:** María Alejandra Gutiérrez Torres, Silvana Ruiz, Karen Morales, Laura Rincon, Frans Serpa, Michelle M. Ahrens, Abul Ariza Manzano, Santiago Callegari.

**Project administration:** María Alejandra Gutiérrez Torres, Santiago Callegari.

**Resources:** María Alejandra Gutiérrez Torres.

**Software:** Santiago Callegari.

**Supervision:** María Alejandra Gutiérrez Torres, Santiago Callegari.

**Validation:** María Alejandra Gutiérrez Torres, Michelle M. Ahrens, Santiago Callegari.

**Visualization:** María Alejandra Gutiérrez Torres, Laura Rincon, Santiago Callegari.

**Writing – original draft:** María Alejandra Gutiérrez Torres, Silvana Ruiz, Karen Morales, Laura Rincon, Frans Serpa, Michelle M. Ahrens, Abul Ariza Manzano, Santiago Callegari, Camila Gómez, Felipe Duran.

**Writing – review & editing:** María Alejandra Gutiérrez Torres, Silvana Ruiz, Karen Morales, Santiago Callegari, Felipe Duran.

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
