## [Decision Letter · Decision Letter 0]

20 May 2024

PGPH-D-24-00677

Disparities of Female Authorship in Colombia A Cross-Sectional Analysis

Dear Dr. Torres,

Thank you for submitting your manuscript to PLOS Global Public Health. After careful consideration, we feel that it has merit but does not fully meet PLOS Global Public Health’s publication criteria as it currently stands. Therefore, we invite you to submit a revised version of the manuscript that addresses the points raised during the review process.

We look forward to receiving your revised manuscript.

Kind regards,

Zahra Zeinali, MD MPH DrGH (c)

Academic Editor

Journal Requirements:

Additional Editor Comments (if provided):

Reviewers' comments:

Reviewer's Responses to Questions

**Comments to the Author**

1. Does this manuscript meet PLOS Global Public Health’s publication criteria ? Is the manuscript technically sound, and do the data support the conclusions? The manuscript must describe methodologically and ethically rigorous research with conclusions that are appropriately drawn based on the data presented.

Reviewer #1: Yes

Reviewer #2: Partly

2. Has the statistical analysis been performed appropriately and rigorously?

Reviewer #1: Yes

Reviewer #2: No

3. Have the authors made all data underlying the findings in their manuscript fully available (please refer to the Data Availability Statement at the start of the manuscript PDF file)?

Reviewer #1: Yes

Reviewer #2: Yes

4. Is the manuscript presented in an intelligible fashion and written in standard English?

Reviewer #1: Yes

Reviewer #2: Yes

5. Review Comments to the Author

Reviewer #1: The research is indeed imperative to be brought to the notice of academia, funders and policymakers about women's under-representation in research contribution. The outcome of this research gives insights into the situation of women's representation and leadership in medical and surgical research. The study is undertaken in the context of Colombia but this could be the situation in most LMICs, and could provide a rationale to undertake similar studies in other LMIC's context. However, answering the following comments would help strengthen your context and methodology:

You have used "women" while citing the literature, whereas you have used the term "female" throughout the manuscript. You might be aware the women is a more inclusive term and female refers to the biological sex at birth. Please justify the use of term "female" for your research or you actually mean women's under-representation in the context of their first authorship in impactful journals.

You mention that "we performed an unadjusted logistic regression to explore the association between the first and last female authors". Please clarity because this sentence does not make it clear about what is the predictor and what are the explanatory variables. Explain the reason to use logistic regression. What hypothesis is the logistic regression testing? How many are research articles, how many are case reports and how many are review articles? Was it not possible to use one single logistic regression and use dummy variables for the type of article? Using dummy variables would give additional insights across these article categories.

Reviewer #2: Referee report on “Disparities of Female Authorship in Colombia: A Cross-Sectional Analysis”

The paper attempts to document underrepresentation of females as first (and last) author(s) of research articles and case studies in medical journals published in Colombia. The author(s) shortlisted journals and articles published in Colombian journals from the year 2018 till 2022. They find that females are less likely to be first (and last) author(s) with considerable heterogeneity across journals depending upon the journal’s sub-specialty.

The research question is really intriguing and worth exploring. I have a handful of comments for the author(s) to think upon.

Major comments:

1. While gender gap in STEM education is well known and a contemporary issue, how and why is looking at the proportion of papers with first female author a way to contextualize gender disparity in medical academia? I am making this point because:

(a) Many journals (and authors) in several disciplines now moved to ordering authors based on the alphabetic order of their names, which makes gender of the first author as random. Further, several journal in a few disciplines (say, economics) also publish a credit author statement – which reflects the contribution of each author irrespective of the order in which the name of the author(s) appear.

Since author(s) of this paper have partly motivated their research question based on the idea that the career progressions of females may be impacted by the first authorship of an individual, major comment 1(a) makes it less of a concern.

1.1 Relatedly, why is last authorship an important outcome variable which the author(s) of this paper have looked upon in their analysis. If author(s) could also motivate why the female last author is an important outcome variable then it may help the reader to appreciate the findings of the paper more.

2. I am not sure if the conclusion written in the Introduction section of the paper follows from the analysis undertaken in the paper.

More precisely, the author(s) state “……..This will serve as the foundation for developing targeted interventions and policy recommendations to promote gender equity in the research ecosystem.

The anticipated results of this study are expected to stimulate collaborative efforts among policymakers, funding agencies, and research institutions to increase female representation and inclusivity in research throughout Latin American nations. By shedding light on the existing disparities and proposing evidence-based solutions, this study seeks to contribute to the creation of a fairer and more equitable academic environment in the region…...”

The claim that documenting of gender disparities can provide foundation for targeted interventions and create a fairer environment seems unlikely. Ideally, if the author(s) through an experiment, or, had undertaken an impact assessment of a policy (using quasi-experimental methods), had shown that certain policies can reduce gender-gaps -- then this conclusion would have followed.

However, in my opinion, the analysis of the paper is only suggestive of (or identifies) a stylized fact that females are less likely to be first authors of research articles and case studies in medical journals published in Colombia. They are not identifying any reasons for this problem or proposing any targeted intervention (that may work) for reducing this through their analysis.

Similarly, conclusions/claims in the discussion section may be revised by the author(s).

Minor points

1. Figure 3 that depicts heterogeneity in the proportion of females first-author based on sub-specialty indicates to me that there is a gender based self-selection in type of sub-specialty choses by individuals. For instance: In comparison to males, females may prefer more to be in sub-specialty jobs like nutrition/dermatology as compared to orthopedics/neurosurgery.

Thus, the lower proportion of females first author in orthopedics/neurosurgery journals may simply be due to lower number females in that sub-specialty (self-selection) and not gender disparity – which is typically perceived as gender discrimination.

Ideally for disparity/discrimination, one should do a relative analysis. For instance: out of the total females in nutrition/dermatology profession, how many females end up being as first authors of published papers. Similarly, out of the total males in nutrition/dermatology profession, how many females end up being as first authors of published papers. Comparison of these two figures can in true sense can serve to provide a sense of disparity in my opinion.

2. I am unable to understand the Logistic regression (Table 2).

Ideally, the logistic regression should have covariates that may explain/predict the likelihood of a female first authorship in a paper. For instance: age of the author, type of journal, seniority of the researcher, number of males co-authors etc. This can serve to provide in true sense the determinants of female first authorship in a research paper. I suggest author(s) to undertake such analysis, if possible. In my opinion, this can add significant value to the paper.

6. PLOS authors have the option to publish the peer review history of their article (what does this mean? ). If published, this will include your full peer review and any attached files.

**Do you want your identity to be public for this peer review?** For information about this choice, including consent withdrawal, please see our Privacy Policy .

Reviewer #1: **Yes: ** Smruti Bulsari

Reviewer #2: No

---

## [Decision Letter · Decision Letter 1]

2 Oct 2024

PGPH-D-24-00677R1

Disparities of Women's Authorship in Colombia: A Cross-Sectional Analysis

Dear Dr. Torres,

Thank you for submitting your manuscript to PLOS Global Public Health. After careful consideration, we feel that it has merit but does not fully meet PLOS Global Public Health’s publication criteria as it currently stands. Therefore, we invite you to submit a revised version of the manuscript that addresses the points raised during the review process.

We look forward to receiving your revised manuscript.

Kind regards,

Zahra Zeinali, MD MPH DrGH (c)

Academic Editor

Journal Requirements:

Additional Editor Comments (if provided):

Reviewers' comments:

Reviewer's Responses to Questions

**Comments to the Author**

1. If the authors have adequately addressed your comments raised in a previous round of review and you feel that this manuscript is now acceptable for publication, you may indicate that here to bypass the “Comments to the Author” section, enter your conflict of interest statement in the “Confidential to Editor” section, and submit your "Accept" recommendation.

Reviewer #1: All comments have been addressed

Reviewer #2: All comments have been addressed

2. Does this manuscript meet PLOS Global Public Health’s publication criteria ? Is the manuscript technically sound, and do the data support the conclusions? The manuscript must describe methodologically and ethically rigorous research with conclusions that are appropriately drawn based on the data presented.

Reviewer #1: Yes

Reviewer #2: Partly

3. Has the statistical analysis been performed appropriately and rigorously?

Reviewer #1: Yes

Reviewer #2: Yes

4. Have the authors made all data underlying the findings in their manuscript fully available (please refer to the Data Availability Statement at the start of the manuscript PDF file)?

Reviewer #1: Yes

Reviewer #2: Yes

5. Is the manuscript presented in an intelligible fashion and written in standard English?

Reviewer #1: Yes

Reviewer #2: Yes

6. Review Comments to the Author

Reviewer #1: The concerns in the original manuscript are addressed satisfactorily. However, following very minor changes are recommended:

1. The authors have mentioned "In a separate analysis, we performed univariate and 2 multivariate logistic regression to ....". It is a common practice to write the numbers in words (e.g. two multivariate logistic regression) in a sentence, when the numbers are in single digit.

2. In the same paragraph, "Lastly, our 2 logistic regression multivariate analysis models...", could be better phrased as "Lastly, both the multivariate logistic regression models...".

3. Table 1 it represents proportion (because the authors have specified the denominators, represented the numbers as fractions and have specified proportions in the brackets below). Instead of "Frequency", the term "Proportion" would be more appropriate. Furthermore, they have mentioned "Women Author" (and "Men Author"). It would be either "Women Authors" (and "Men Authors") or "Woman Author" (and "Man Author").

I would like the authors to know that the gender disparity in surgical versus non-surgical disciplines - both about representativeness of women and first authorship is very nicely discussed.

Reviewer #2: I think the author(s) have thoroughly revised the paper and I appreciate their efforts. However, I just have 3 minor points to make:

1. Contribution to the literature: I still feel the author(s) state more than what the paper actually contributes. This is especially true when the author(s) write:

“This study seeks to address these challenges by generating evidence-based recommendations for policy and practice, given the insufficient data regarding the scope and severity of gender inequality in scientific fields at the regional level.”

I doubt if any evidence-based recommendations are generated for policy and practice. Rather, I feel it’s a stylized fact which the authors are highlighting about the gender inequality in Medical research.

2. Descriptive Statistics: Author(s) state that “women’s first authorships increased slightly in 2019 and 2022 but decreased in 2021. Similarly, last women’s authorships increased in 2021 but decreased in 2019 and 2022, indicating a possible inverse relationship, as illustrated in Figure 2.”

To the naked eye, these differences are very small and it hard to conclude whether actually or not these point estimates are statistically different from one another. In order to make this claim concretely, we need confidence intervals attached to each year's mean value.

Logistic regression: While undertaking the logistic regression the author(s) state that “When assessing factors associated with women being the last authors, the variables positively associated with this were the first author being a woman and the year of publication 2021”

How can year of paper of publication be concluded as a determinant of gender disparity. Rather, the year of publication just represents the trend. Hence, in 2021, it was more likely that to have women as last authors compared to the base/reference year (2018). In my opinion, year of publication cannot be interpreted as a covariate determining the gender inequality.

7. PLOS authors have the option to publish the peer review history of their article (what does this mean? ). If published, this will include your full peer review and any attached files.

**Do you want your identity to be public for this peer review?** For information about this choice, including consent withdrawal, please see our Privacy Policy .

Reviewer #1: **Yes: ** Smruti Bulsari

Reviewer #2: No

---

## [Editor Report · Decision Letter 2]

26 Nov 2024

Disparities of Women's Authorship in Colombia: A Cross-Sectional Analysis

PGPH-D-24-00677R2

Dear Dr. Torres,

We are pleased to inform you that your manuscript 'Disparities of Women's Authorship in Colombia: A Cross-Sectional Analysis' has been provisionally accepted for publication in PLOS Global Public Health.

Best regards,

Zahra Zeinali, MD MPH DrGH (c)

Academic Editor